# Development of Flow Cytometric Assay for Detecting Papillary Thyroid Carcinoma Related hsa-miR-146b-5p through Toehold-Mediated Strand Displacement Reaction on Magnetic Beads

**DOI:** 10.3390/molecules26061628

**Published:** 2021-03-15

**Authors:** Yue Wu, Jiaxue Gao, Jia Wei, Jingjing Zhou, Xianying Meng, Zhenxin Wang

**Affiliations:** 1Department of Thyroid Surgery, The First Hospital of Jilin University, Changchun 130021, China; wuyuejlu@163.com (Y.W.); weijia@ciac.ac.cn (J.W.); zzzj4679@163.com (J.Z.); 2State Key Laboratory of Electroanalytical Chemistry, Changchun Institute of Applied Chemistry, Chinese Academy of Sciences, Changchun 130022, China; gjx@ciac.ac.cn

**Keywords:** hsa-miR-146b-5p, magnetic beads, flow cytometry, thyroid carcinoma

## Abstract

In this work, a simple enzyme-free flow cytometric assay (termed as TSDR-based flow cytometric assay) has been developed for the detection of papillary thyroid carcinoma (PTC)-related microRNA (miRNA), hsa-miR-146b-5p with high performance through the toehold-mediated strand displacement reaction (TSDR) on magnetic beads (MBs). The complementary single-stranded DNA (ssDNA) probe of hsa-miR-146b-5p was first immobilized on the surface of MB, which can partly hybridize with the carboxy-fluorescein (FAM)-modified ssDNA, resulting in strong fluorescence emission. In the presence of hsa-miR-146b-5p, the TSDR is trigged, and the FAM-modified ssDNA is released form the MB surface due to the formation of DNA/RNA heteroduplexes on the MB surface. The fluorescence emission change of MBs can be easily read by flow cytometry and is strongly dependent on the concentration of hsa-miR-146b-5p. Under optimal conditions, the TSDR-based flow cytometric assay exhibits good specificity, a wide linear range from 5 to 5000 pM and a relatively low detection limit (LOD, 3σ) of 4.21 pM. Moreover, the practicability of the assay was demonstrated by the analysis of hsa-miR-146b-5p amounts in different PTC cells and clinical PTC tissues.

## 1. Introduction

MicroRNAs (miRNAs) are single-stranded non-coding RNAs, ranging in length from 19 to 25 nucleotides, which were first unintentionally discovered in *Caenorhabditis elegans* by Lee and colleagues in 1993 [1]. MiRNAs enable regulating the expression of target genes by binding to target messenger RNA (mRNA) and inducing its degradation or translation inhibition [2]. Consequently, miRNAs are involved in the regulation of various biological processes in organisms, such as cell proliferation, differentiation, metabolism, embryogenesis, inflammation, senescence and programmed cell death [3,4]. Recently, massive studies have demonstrated that the abnormal expression of miRNA is closely related to the occurrence and development of human malignant tumors including gastric cancer, liver cancer, breast cancer and thyroid cancer [5]. There is a rapidly increasing demand for miRNA detection [6,7], because miRNA is considered as a novel biomarker for cancer diagnosis, prognostic analysis and molecular targeted drug development. Due to their inherent characteristics including short sequence, low abundance and high sequence similarity among family members, it is difficult to precisely detect and analyze miRNAs in practical samples. Recently, various assays/methods have been developed for the detection of miRNAs, such as Northern blot [8], microarray-based method [9], reverse transcription polymerase chain reaction (RT-PCR) [10], isothermal exponential amplification method (EXPAR) [11] and rolling-circle amplification (RCA) method [12]. Although most of these assays/methods can be employed to detect miRNAs with high sensitivity and accuracy, they have some drawbacks/limitations, such as the complex design of probe/primer/template sequence, tedious detection steps, long assaying time and requirement of expensive reagents (e.g., enzymes). Therefore, there is still a strong desire to develop a feasible approach with high simplicity for rapid quantification of miRNAs in practical samples.

As a versatile tool, magnetic beads (MBs) have been extensively employed for the purification and quantitative detection of different analytes in complex biological matrices since MBs have several unique advantages including good stability, uniform size distribution, easy functionalization and rapid response to applied magnetic field [13,14,15]. For instance, Li and colleagues have developed a flow cytometric bead assay for the simultaneous detection of multiple miRNAs through the integration of size-coded MBs with a two-step enzyme-mediated cascading signal amplification [16]. Toehold-mediated strand displacement reaction (TSDR) generally takes place in the case of partially hybridized duplex with an overhanging single-stranded toehold domain containing 5−8 nucleotides, in which the displacement reaction is followed by the formation of the toehold-target duplex [17]. The TSDR is a useful strategy for the construction of analytical assays with good performance because the displacement process is fast, predictable and easy to differentiate the mismatched sequences [18,19,20,21,22,23]. In addition, MB-assisted TSDR with different detection principles including fluorescent [24,25], photoelectrochemical [22], colorimetric and chemiluminescent [26,27] have been developed for the detection of cancer-related miRNAs.

As the most common type of thyroid cancer, papillary thyroid carcinoma (PTC) has been diagnosed with increasing frequency in recent decades in many developed countries and China. Although up to 90% of patients with PTC at early stage can achieve long-term (more than 5 years) survival, life-long surveillance is required [28,29,30]. Because of non-total thyroidectomy, the presence of anti-thyroglobulin (Tg) antibodies, and/or lack of iodine avidity, the clinical used gold standard of life-long surveillance of PTC, monitoring serum thyroglobulin (Tg) levels is not suitable for up to 25% of patients with PTC [31]. It has been demonstrated that the occurrence and development of PTC has strong association with high levels of several miRNAs including miR-146b, miR-222, and miR-221 [32,33,34,35]. For instance, the family member of miR-146b, hsa-miR-146b-5p exhibits very high expression level in PTC, which can promote the proliferation, migration, invasion and cell cycle progression of PTC cells through the regulation of cell signaling pathways including downregulating the expression of CCDC6 [36], IRAK1 and other PTC-related genes [37,38]. Therefore, hsa-miR-146b-5p can be used as a potential biomarker for the diagnosis of PTC and help us to understand the mechanism of tumor development.

Herein, we proposed an enzyme-free flow cytometric assay (termed as a TSDR-based flow cytometric assay) for the detection of thyroid cancer-associated miRNA, hsa-miR-146b-5p through the combination of MB-based TSDR and flow cytometry fluorescence detection. Taking advantage of the flow cytometry’s strong analysis ability (such as high sensitivity and high throughput) and a MB’s magnetic separation capacity, the as-proposed flow cytometric assay exhibits high performance compared to other methods, which can be employed to rapidly monitor hsa-miR-146b-5p levels in practical samples including cell lysates and the clinical tissue homogenate of PTC, showing great potential in clinical diagnosis.

## 2. Materials and Methods

### 2.1. Materials and Reagents

Oligonucleotides (see Table 1 for details) were synthesized by Sangon Ltd. Co. (Shanghai, China). Dynabeads^®^ M-270 streptavidin modified (M-270 MBs, 2.8 μm) and 5 × banding and washing (B&W) buffer (25 mM Tris-Hcl, 2.5 mM EDTA, 5 M NaCl, pH 7.5) were obtained from Thermo Fisher Scientific Co. (Asheville, NC, USA). Diethylpyrocarbonate-treated distilled water (DEPC water) was supplied by Dingguo Biotechnology Ltd. (Beijing, China). All buffers were prepared with DEPC water to prevent miRNA degradation. Dulbecco’s modified Eagle’s medium (DMEM), RPMI-1640 medium and fetal bovine serum (FBS) were purchased from HyClone Co. (Los Angeles, CA, USA). Other used reagents were analytical grade, and were purchased from Sinopharmaceutical Reagents Ltd. (Shanghai, China).

### 2.2. Cell Culture and Tissue Sample Collection

Human thyroid cancer cell lines (TPC-1, K1 and C643) and human normal thyroid cell lines (Nthy-ori 3-1) were purchased from Shanghai Cell Bank, Chinese Academy of Sciences (Shanghai, China). All cells were cultured at 5% CO_2_ at 37 °C in a humidified incubator (Thermo Co., Asheville, NC, USA). The PTC cell lines (K1 and TPC-1) cells were cultured in DMEM supplemented with 10% FBS and 1% penicillin–streptomycin, while the undifferentiated thyroid carcinoma cell lines (C643) and human normal thyroid cell line (Nthy-ori 3-1) were cultured in RPMI-1640 supplemented with 10% FBS and 1% penicillin–streptomycin, respectively. After full growth, the cells were digested with trypsin and counted with a Dakewe mini cell counter (Dakewe Biotech Ltd., Shenzhen, China).

Thyroid tissue collection for this work was approved by the Ethics Committee of Jilin University, and all subjects had signed informed consent prior to participating in the study. Samples were collected from 16 patients with PTC and 16 patients with nodular goiter (NG) who underwent thyroid surgery at the Thyroid Surgery department of Bethune First Hospital, Jilin University from June to July 2020, respectively. The diagnosis of each case was independently confirmed by two pathologists according to WHO classification (see Table 2 for patient details). The clinical stages were classified according to the American Joint Committee on Cancer (AJCC) tumor-lymph node metastasis (TNM) classification system. The as-obtained tissue samples were immediately stored at −80 °C until further use.

### 2.3. The Extraction of miRNAs

According to the manufacturer’s instruction, the total RNAs of 1 mL cell solution (1 × 10^6^ cells/mL) were extracted by the commercially available miRNA extraction kit (Qiagen Co., Inc., New York, NY, USA) in the ultra-clean platform. The extracted total RNAs were dispersed in 50 μL DEPC water. Thirty milligrams (30 mg) of thyroid pathological tissue samples were crushed and homogenized. The total RNAs of pre-treated tissue sample were extracted by the miRNA extraction kit in ultra-clean platform and re-dispersed in 50 μL DEPC water.

### 2.4. Preparation of MB-Probe Conjugates

The 0.2 mg M-270 MBs were resuspended in 1 mL 1 × B&W buffer, and washed three times to remove the passivator and preservative from the surface of MBs. The MBs were then collected under a magnetic frame (AMD06, Almedton Ltd., Shenzhen, China), and resuspended in 400 μL 2 × B&W buffer. Moreover, 400 μL p-DNA in various concentrations of distilled water was mixed with the MBs suspension, and incubated under gentle shaking (170 rpm) at 25 °C for 30 min. The as-obtained products (MB@ssDNA) were washed with 1 mL 1 × B&W buffer (three times) and resuspended in 400 μL reaction buffer (20 mM Tris-HCl, 150 mM NaCl, 15 mM MgCl_2_, pH 7.0). In addition, 4 μL f-DNA (100 μM) was added into the MB@ssDNA suspension, incubated under gentle shaking (170 rpm) at 30 °C for 1 h, and washed with 1 mL reaction buffer. The final product (MB@dsDNA) was dispersed in 2 mL reaction solution for further use. The fluorescence intensity of MB@dsDNA was read by a BD ACCURI C6 flow cytometer (BD Co., New York, NY, USA).

### 2.5. Detection of hsa-miR-146b-5p

Five microliters hsa-miR-146b-5p in various concentrations were added into 45 μL MB@dsDNA, incubated under gentle shaking (170 rpm) at 37 °C for 1 h, and directly read by flow cytometry. In total, 10,000 MBs were recorded, and the FL1-A mean fluorescence intensity (MFI) of the MB was used for the quantitative analysis of hsa-miR-146b-5p. The content of hsa-miR-146b-5p in the tested sample was analyzed by calculating the difference value (ΔF) between the MFI of the tested sample and the blank sample. For the detection of hsa-miR-146b-5p in practical samples, 5 μL RNA extracts were added into 45 μL MB@dsDNA, incubated and detected as previously described.

## 3. Results

### 3.1. Principle of the TSDR-Based Flow Cytometric Assay

Scheme 1 shows the principle of TSDR-based flow cytometric assay for detection of PTC-related miRNA, hsa-miR-146b-5p through the TSDR on MBs. In this case, the biotinylated probe ssDNA (p-DNA) was conjugated on the surface of the streptavidin functionalized MBs by the strong interaction of biotin with streptavidin. The carboxy-fluorescein (FAM)-labeled ssDNA (f-DNA) was then hybridized with p-DNA to prepare the fluorescent MB probe (MB@dsDNA). The MB@dsDNA is readily read by flow cytometry. After hybridization, an exposed toehold of six bases length was formed at 5′-end of p-DNA. Based on literature reports [17,18,19,20,21,22,23], a longer (>8 bases) toehold region will cause the instability of the hybridization of f-DNA and p-DNA, resulting in a poor detection specificity, while shorter (<5 bases) toehold region will cause a decrease in the reaction rate, resulting in a low detection sensitivity. Therefore, the toehold of six bases was used in our experiment. In the presence of hsa-miR-146b-5p, p-DNA is hybridized with hsa-miR-146b-5p through the exposed toehold region, and TSDR is initiated. The process leads to disassociate f-DNA from the surface of MBs, resulting in the decrease in the fluorescence signals of MB@dsDNA. The change of fluorescence signal of MB@dsDNA (ΔF = F_0_ − F, here, F_0_ is the fluorescence intensity of MB@dsDNA, while F is the fluorescence intensity of MB@dsDNA after interaction with a certain amount of hsa-miR-146b-5p) is negatively dependent on the concentration of hsa-miR-146b-5p. For obtaining high detection accuracy, the average fluorescence intensity (MFI) of 10,000 MB samples was used to evaluate the concentration of hsa-miR-146b-5p.

### 3.2. Optimization of the Experimental Conditions

In order to obtain the high detection performance of hsa-miR-146b-5p, several experimental conditions were optimized including the concentration of p-DNA on the surface of MB, the reaction temperature and reaction time of MB@dsDNA with hsa-miR-146b-5p. It known that the detection efficiency of TSDR-based assays largely depend on the initial concentration of p-DNA on the surface of MBs. As shown in Figure 1, the fluorescence signal (F_0_) increased with the increase in p-DNA concentration, while the concentration of MB was kept constant. However, the detection sensitivity (ΔF/F_0_) was decreased when the concentration of p-DNA is higher than 1 nM. In order to obtain the ideal dynamic range and the sensitivity of hsa-miR-146b-5p detection, 1 nM p-DNA was selected for the preparation of MB@dsDNA. DNA hybridization efficiency and miRNA replacement efficiency are strongly affected by reaction temperature. As shown in Figure 2, ΔF was increased by increasing the reaction temperature in the range of 27–37 °C, and obtain saturation when the reaction temperature was higher than 37 °C. Thus, 37 °C was selected as the optimal reaction temperature. To further increase assay performance, the reaction time was also optimized. As shown in Figure 3, the highest ΔF was obtained when the MB@dsDNA was reacted with hsa-miR-146b-5p at 37 °C for 60 min. Therefore, in the following experiments, the MB@dsDNA were prepared by the reaction of 0.2 mg/mL MBs with 1 nM p-DNA, and the MB@dsDNA were reacted with hsa-miR-146b-5p at 37 °C for 60 min.

### 3.3. Detection Performance of the TSDR-Based Flow Cytometric Assay

Under optimal reaction conditions, the fluorescence intensity of MB@dsDNA is decreased by increasing the concentration of hsa-miR-146b-5p (as shown in Figure 4a). It is consistent with the fact that the more hsa-miR-146b-5p exists in the solution, the more f-DNA will be dissociated from the MB@dsDNA. As shown in Figure 4b, a linear relationship between ΔF and the logarithm values of hsa-miR-146b-5p concentrations in the range of 5 pM to 5 nM is obtained. The detection limit (LOD) is estimated to be 4.21 pM according to the 3σ/S rule (σ is the standard deviation (*n* = 3) for the blank solution, and S is the slope of the calibration curve), which is comparable and/or better than those reported in the literature [8,39,40,41].

### 3.4. Specificity of the TSDR-Based Flow Cytometric Assay

To address the specificity of TSDR-based flow cytometric assay, four miRNAs were used as interferences. The hsa-miR-146a-5p has similar sequence with the hsa-miR-146b-5p except for one base difference (A to G at no. 18 site). Hsa-miR-21, hsa-miR-221, and hsa-miR-222 are associated with the occurrence and development of thyroid cancer [35]. As shown in Figure 5a, the TSDR-based flow cytometric assay shows negligible ΔF towards four interferences. The experimental result indicates that the TSDR-based flow cytometric assay has good specificity.

### 3.5. Detection of Intracellular hsa-miR-146b-5p

To demonstrate its capability, the TSDR-based flow cytometric assay was applied to the profile activity levels of hsa-miR-146b-5p in different cells including two PTC cells (TPC-1 and K1), one undifferentiated thyroid cancer cell (C643) and one normal thyroid cell (Nthy-ori 3-1). The expression levels of cellular hsa-miR-146b-5p exhibit significant difference among different cells (as shown in Figure 5b), and follow the order, TPC1 ≅ K1 > C643 > Nthy-ori 3-1, which is consistent with literature-reported results [42,43], i.e., PTC cells express a higher level of hsa-miR-146b-5p than that of undifferentiated thyroid cancer cell, and normal thyroid cell expresses the lowest level of hsa-miR-146b-5p.

### 3.6. Detection of hsa-miR-146b-5p in Clinical Tissue Samples

For further demonstrating its applicability, the TSDR-based flow cytometric assay was used to explore the expression differences of hsa-miR-146b-5p in tissue samples from patients with PTC and/or benign thyroid lesions (NGs). In this case, 16 specimens from PTC patients and 16 specimens from NG patients were collected through the surgery. The relevant information of the patients was presented in Table 2. As shown in Figure 6a, the expression levels of hsa-miR-146b-5p in PTC tissues were significantly higher than those in NG tissues and the difference was statistically significant (*p* < 0.001), which is consistent with the reported results of relevant studies [34]. In order to prove the accuracy of this method, we drew a receiver operating characteristic curve (ROC) based on this result. The area under the ROC curve (AUC) was calculated to be 0.87 (as shown in Figure 6b). The sensitivity and specificity for the diagnosis of PTC are 81.3% and 75%, when the ΔF value is 639.01. The above results show that this method has good accuracy and practicability and has the potential to be applied in clinical practice for discriminating benign from malignant tumors. For comparison, the expression levels of hsa-miR-146b-5p in PTC tissues and NG tissues were also analyzed by qPCR method (as shown in the Figure 7). The result of TSDR-based flow cytometric assay is highly consistent with that of the qPCR analysis (as shown in Figure 8). These results demonstrate that the TSDR-based flow cytometric assay has great promise as a practical approach that can provide high accuracy.

## 4. Conclusions

In summary, we developed an enzyme-free TSDR-based flow cytometric assay for the rapid, sensitive and selective detection of PTC-related miRNA hsa-miR-146b-5p based on the TSDR on MBs. The enzyme-free manner allows the convenient and robust detection of target miRNA with low cost. In combination with the high throughput characteristic of flow cytometry, the detection can be accomplished within 2 h except for sample preparation. In particular, the TSDR-based flow cytometric assay has been successfully employed to detect the content of hsa-miR-146b-5p in various cell lines and clinical pathological samples of PTC tissues and NG tissues. Although only hsa-miR-146b-5p is detected in the proof of principle experiment, this study provides a powerful tool for evaluating various miRNAs levels in complex matrices though changing the TSDR. Therefore, an as-proposed TSDR-based flow cytometric assay could be used as a valuable adjunctive method for improving the diagnosis accuracy of PTC because several miRNAs are significantly unregulated in PTC. In addition, with the help of existing nucleic acid enrichment techniques, our approach could be used to detect miRNAs in body fluids, which exhibits great potential in biomedical applications such as the noninvasive diagnosis of cancer.

## Data Availability

The data presented in this study are available on request from the corresponding author.

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
