# Peer review of "Development of Flow Cytometric Assay for Detecting Papillary Thyroid Carcinoma Related hsa-miR-146b-5p through Toehold-Mediated Strand Displacement Reaction on Magnetic Beads"

_molecules, 2021, doi:10.3390/molecules26061628_

Round 1

Reviewer 1 Report

In this paper entitled Development of flow cytometric assay for detecting papillary thyroid carcinoma related hsa-miR-146b-5p through toehold-mediated strand displacement reaction on magnetic beads, Wu and colleagues develop an enzyme-free and straightforward method to detect miRNA in biological samples.

The paper is exciting. Data are presented precisely. Conclusions are highly significant regarding the simplicity, low cost, and precision of the method's purpose. 

The article is very well written and clearly explained. References are adequate. The questions addressed in the paper fit very well to the field of study. Conclusions and possible application that emerges from this work are relevant in the possible diagnostic use of miRNAs in the clinic with an easy-to-implement method.

Since this work has higher novelty and has substantial physiological relevance, some specific points need to be addressed to improve clarity and relevance to broad readers interested in this technology in other fields of application.

First, a further expanded explanation of the process to select adequate f-DNA sequences and a clear discussion about the importance of adequate complementary sequences between 5' end in fDNA and miRNA could help understand the TSDR methods. In this sense, in Schene 1, a more detailed graphic explanation of MBdsDNA preparation should help.

Second, a more extensive discussion section is needed to discuss the results, their novelty regarding the previous enzymatic method to detect miRNA, and the possible application in liquid biopsies. That improvement should be of interest from the most audience in search of new noninvasive methods to determine new clinical relevant biomarkers.  

Regarding the sensitivity of the method, data are clearly and presented. However, It has some queries to be answered. In figure 4b, miRNA concentration, regarding ΔF value, became linear at ΔF values close to 300. However, in figure 5. the determination of miRNA in cell lines shown ΔF values lower than 200. Even in the same figure, panel A authors sentence as "negative" (For miR-146a, for instance) ΔF values close to 200, similarly levels detected from Nthy-ori 3-1 cell line. These confusing data need to be discussed in detail in line with the possible "physiological" concentration of miRNA in tissue samples vs. plasma sample and cell line supernatant.

Another minor change that could improve clarity is to put into the titles and experimental conditions of the panels in the figures. for example, in histograms from figures 2-4

Author Response

  1. First, a further expanded explanation of the process to select adequate f-DNA sequences and a clear discussion about the importance of adequate complementary sequences between 5' end in fDNA and miRNA could help understand the TSDR methods. In this sense, in Schene 1, a more detailed graphic explanation of MB dsDNA preparation should help.

Response: Thank you for your suggestion. We have re-drawn the experimental schematic diagram in more detail according to the suggestion. Selecting the adequate f-DNA sequences is of great significance to the efficiency and stability of TSDR reactions. Toehold region is the complementary defect region of f-DNA and p-DNA, hsa-miR146b-5p binds to the p-DNA through this region. Although shorter f-DNA can form longer toehold region, this will cause the instability of the hybridization of f-DNA and p-DNA, resulting in a poor detection specificity. Longer f-DNA can form shorter toehold region, but it will cause a decrease of the reaction rate. Based on the literature reports, (refs. 17-23), the toehold region with 5 to 8 bases in length is the most appropriate. Therefore, the toehold region of 6 base length was used in our assay. Sentences ‘Based on literature reports [17-23], longer (>8 bases) toehold region will cause the instability of the hybridization of f-DNA and p-DNA, resulting in a poor detection specificity, while shorter (<5 bases) toehold region will cause a decrease of the reaction rate, resulting in a low detection sensitivity. Therefore, toehold of 6 bases was used in our experiment.’ have been added in the page 5 of revised manuscript for addressing this matter.

  1. Second, a more extensive discussion section is needed to discuss the results, their novelty regarding the previous enzymatic method to detect miRNA, and the possible application in liquid biopsies. That improvement should be of interest from the most audience in search of new noninvasive methods to determine new clinical relevant biomarkers.  

Response: Thank you for your suggestion. Sentences ‘The enzyme-free manner allows the convenient and robust detection of target miRNA with low-cost. Combination with the high throughput characteristic of flow cytometry, the detection can be accomplished within 2 h except for sample preparation.’ and ‘With the help of existing nucleic acid enrichment techniques, our approach could be used to detect miRNAs in body fluids, which exhibits great potential in biomedical applications such as noninvasive diagnosis of cancer.’ have been added in the revised manuscript for addressing this matter.

  1. Regarding the sensitivity of the method, data are clearly and presented. However, It has some queries to be answered. In figure 4b, miRNA concentration, regarding ΔF value, became linear at ΔF values close to 300. However, in figure 5. the determination of miRNA in cell lines shown ΔF values lower than 200. Even in the same figure, panel A authors sentence as "negative" (For miR-146a, for instance) ΔF values close to 200, similarly levels detected from Nthy-ori 3-1 cell line. These confusing data need to be discussed in detail in line with the possible "physiological" concentration of miRNA in tissue samples vs. plasma sample and cell line supernata?

Response: We think that there is a misunderstanding. Normally, the lowest value of the linear range is usually higher than the lowest detectable value. The lowest detectable value is dependent on the sensitivity of used instrument and the background of sensing system. In our experiment, the lowest detectable value of ΔF is about 100. However, lowest detectable value is meaningless if it is lower than detection limit (LOD) of assay. According to the 3σ/S rule (σ is the standard deviation (n=3), the LOD of our assay is 4.21 pM, which cprresponds the value of ΔF is about 250.

  1. The English should be polished.

Response: The English writing of revised manuscript has been improved.

Reviewer 2 Report

The manuscript by Wu et al. deals with an important topic of devising a new assay for miRNA detection, using real life specimens collected from the patients. 

However, the study design is not completely appropriate, since there is no control evaluation performed by some of the standard miRNA detection procedures such as Northern blotting, (qRT-PCR), sequencing or microarray-based hybridization. 

In addition, it is not very convincing from the presented data that the change of fluorescence is proportional to the concentration of the miRNA... 

The manuscript is full of spelling mistakes, including very often wrongly written name of the miRNA in question. Extensive editing of English language and style is required. 

More details available in the attached pdf with comments. 

Reviewer 3 Report

This interesting article presents a new flow-based assay to detect a specific microRNA subtype, hsa-miR-146b-5p, increased in papillary thyroid carcinoma. Although the technique is nicely described, there some major drawbacks of the technique that should especially be addressed in the conclusions. Also statistical analysis needs improvements.

„Because of nontotal thyroidectomy, presence of anti-thyroglobulin (Tg) antibodies, and/or lack of radioactive,“ here seems to be an error and the authors mean probably lack of uptake of radioactive iodine???

„Take advantages of the flow cytometry's strong analysis ability and MB’s magnetic separation capacity“ please be more specific. What does strong analysis ability means? High sensitivity? Clinical application is easy?

The statistical tests used in figures 5 and 6 may not be adequately chosen: one-sided paired student t-test. Are the results shown also significant with a two-sided unpaired t-test or non-parametric testing?

Conclusion is very short and looks like a summary and NOT a conclusion. An huge effort should be made to improve it. Here one would expect some reflections on the use of this technique in clinical use and to show the advantages and drawbacks of this technique. What is it’s value?

For example: How could this technique help the clinicians in to increase the cited 25% of patients that are missed with the current gold standard TG in the cancer follow-up? (Reference 31 of the authors).

One major drawback of this paper is that no  data on blood-derived Mir is shown. It is nice to see that the assay works for tissue samples and cell cultures, but I would see it’s use especially as an addition of current techniques. There is no much clinical utility to check the tumor tissue.

Some language issues, such as:

„which can promotes the proliferation“

„Take advantages of the flow cytometry's strong analysis ability and MB’s magnetic separation capacity, the as-proposed flow cytometric assay exhibits high performance compared to other methods, which can be employed to rapidly monitor hsa-miR-146b-5p levels in practical samples including cell lysates and clinical tissue homogenate of PTC, showing great potential in clinical diagnosis.“

“This study provides a powful tool”

Author Response

  1. Because of nontotal thyroidectomy, presence of anti-thyroglobulin (Tg) antibodies, and/or lack of radioactive,“ here seems to be an error and the authors mean probably lack of uptake of radioactive iodine???

Response: Thank you for your comment. We mean that some malignant tumors are insensitive to radioactive iodine treatment because they exhibit poor affinity with iodine. And we change the sentence into “Because of nontotal thyroidectomy, presence of anti-thyroglobulin (Tg) antibodies, and/or lack of iodine avidity” in the revised manuscript.

2.Take advantages of the flow cytometry's strong analysis ability and MB’s magnetic separation capacity“ please be more specific. What does strong analysis ability means? High sensitivity? Clinical application is easy?

Response: Thank you for your comment. We revised the sentence as “Taking advantage of the flow cytometry's strong analysis ability (such as high sensitivity and high throughput) and MB’s magnetic separation capacity” for addressing this matter. Yes, the flow cytometry is extensively used in clinical laboratory.

  1. The statistical tests used in figures 5 and 6 may not be adequately chosen: one-sided paired student t-test. Are the results shown also significant with a two-sided unpaired t-test or non-parametric testing?

Response: We have carefully checked our statistical methods, and we used two-sided unpaired Student’s t-test to analyze the data. The caption of Figure 5 and 6 were revised.

4.Conclusion is very short and looks like a summary and NOT a conclusion. An huge effort should be made to improve it. Here one would expect some reflections on the use of this technique in clinical use and to show the advantages and drawbacks of this technique. What is it’s value?

For example: How could this technique help the clinicians in to increase the cited 25% of patients that are missed with the current gold standard TG in the cancer follow-up? (Reference 31 of the authors).

One major drawback of this paper is that no  data on blood-derived Mir is shown. It is nice to see that the assay works for tissue samples and cell cultures, but I would see it’s use especially as an addition of current techniques. There is no much clinical utility to check the tumor tissue.

Response: Thank you for your suggestion. In this proof-in-principle experiment, the TSDR-based flow cytometric assay was not employed to detect miRNA in blood because the abundance of miRNA hsa-miR-146b-5p in blood is extremely low. The as-proposed TSDR-based flow cytometric assay could be used as valuable adjunctive method for improve diagnosis accuracy of PTC because hsa-miR-146b-5p is unregulated significantly in PTC. Based on the reviewer’s, we revised the conclusion as ‘In summary, we developed a enzyme-free TSDR-based flow cytometric assay for rapid, sensitive and selective detection of PTC-related miRNA hsa-miR-146b-5p based on the TSDR on MBs. The enzyme-free manner allows the convenient and robust detection of target miRNA with low-cost. Combination with the high throughput characteristic of flow cytometry, the detection can be accomplished within 2 h except for sample preparation. In particular, the TSDR-based flow cytometric assay has been successfully employed to detect the content of hsa-miR-146b-5p in various cell lines and clinical pathological samples of PTC tissues and NG tissues. Although only hsa-miR-146b-5p is detected in the proof of princple experiment, this study provides a powerful tool for evalating various miRNAs levels in complex matrices though changing the TSDR. Therefore, as-proposed TSDR-based flow cytometric assay could be used as valuable adjunctive method for improve diagnosis accuracy of PTC because several miRNAs are unregulated significantly in PTC. In addition, with the help of existing nucleic acid enrichment techniques, our approach could be used to detect miRNAs in body fluids, which exhibits great potential in biomedical applications such as noninvasive diagnosis of cancer.’.

  1. The English should be polished.

Response: The English writing of revised manuscript has been improved.

Round 2

Reviewer 2 Report

It is not clear why the PCR results are presented for reviewing only and not included in the final manuscript

Author Response

Thank you for your suggestion. The mentioned figures were added in the revised manuscript.

Reviewer 3 Report

Thank you for addressing all the questions and comments. I agree with the other reviewer that the additional graphs should absolutely included in the manuscript and should not be withhold.

Author Response

(The authors gave the same response as above.)
